

# Association of pre-pregnancy body mass index with adverse pregnancy outcome among first-time mothers

Li Li[1,2,3,*], Yanhong Chen[1,3,*], Zhifeng Lin[5], Weiyan Lin[5], Yangqi Liu[5], Weilin Ou[5], Chengli Zeng[5] and Li Ke[1,4]

[1] Department of Obstetrics and Gynecology, The Third Affiliated Hospital of Guangzhou Medical University, Guangzhou, Guangdong, China
[2] Center for Reproductive Medicine/ Department of Fetal Medicine and Prenatal Diagnosis/BioResource Research Center, The Third Affiliated Hospital of Guangzhou Medical University, Guangzhou, Guangdong, China
[3] Key Laboratory for Major Obstetric Diseases of Guangdong Provinces, Guangzhou, Guangdong, China
[4] Key Laboratory of Reproduction and Genetics of Guangdong Higher Education Institutes, Guangzhou, Guangdong, China
[5] Department of Medical Record, The Third Affiliated Hospital of Guangzhou Medical University, Guangzhou, Guangdong, China
* These authors contributed equally to this work.

Corresponding author
Li Ke, keli1221@126.com

## ABSTRACT

**Background**. Studies have reported an increased risk of adverse pregnancy outcome associated with pre-pregnancy body mass index (BMI). However, the data on such associations in urban areas of southern Chinese women is limited, which drive us to clarify the associations of pre-pregnancy BMI and the risks of adverse pregnancy outcomes (preterm birth (PTB) and low birth weight (LBW)) and maternal health outcomes (gestational hypertension and cesarean delivery).

**Methods**. We performed a hospital-based case-control study including 3,864 Southern Chinese women who gave first birth to a live singleton infant from January 2015 to December 2015. PTB was stratified into three subgroups according to gestational age (extremely PTB, very PTB and moderate PTB). Besides, we combined birth weight and gestational age to dichotomise as being small for gestational age (SGA, less than the tenth percentile of weight for gestation) and non-small for gestational age (NSGA, large than the tenth percentile of weight for gestation), gestational week was also classified into categories of term, 34-36 week and below 34 week.. We then divided newborns into six groups: (1) term and NSGA; (2) 34–36 week gestation and NSGA; (3) below 34 week gestation and NSGA; (4) term and SAG; (5) 34–36 week gestation and SAG; (6) below 34 week gestation and SAG. Adjusted logistic regression models was used to estimate the odds ratios of adverse outcomes.

**Results**. Underweight women were more likely to give LBW (AOR = 1.44, 95% CI [1.11–1.89]), the similar result was seen in term and SAG as compared with term and NSAG (AOR = 1.78, 95% CI [1.45–2.17]), whereas underweight was significantly associated with a lower risk of gestational hypertension (AOR = 0.45, 95% CI [0.25–0.82]) and caesarean delivery (AOR = 0.74, 95% CI [0.62–0.90]). The risk of extremely PTB is relatively higher among overweight and obese mothers in a subgroup analysis of PTB (AOR = 8.12, 95% CI [1.11–59.44]; AOR = 15.06, 95% CI [1.32–172.13], respectively). Both maternal overweight and obesity were associated with a greater risk

of gestational hypertension (AOR = 1.71, 95% CI [1.06–2.77]; AOR = 5.54, 95% CI [3.02–10.17], respectively) and caesarean delivery (AOR = 1.91, 95% CI [1.53–2.38]; AOR = 1.85, 95% CI [1.21–2.82], respectively).

**Conclusions**. Our study suggested that maternal overweight and obesity were associated with a significantly higher risk of gestational hypertension, caesarean delivery and extremely PTB. Underweight was correlated with an increased risk of LBW and conferred a protective effect regarding the risk for gestational hypertension and caesarean delivery for the first-time mothers among Southern Chinese.

## INTRODUCTION

Preterm birth (PTB) is an important adverse pregnancy outcome with a significant impact on infant mortality and morbidity (*Boghossian et al., 2016*). The incidence of PTB worldwide is expected to be 11.1%, and China has more than 1.1 million PTB each year, which ranks second in the world (*Blencowe et al., 2012*). In spite of high-level advancement in healthcare services, the rate of PTB still seems to be climbing. According to statistics, perinatal mortality is as high as 70% in low birth weight (LBW) with neonatal weight below 2,500 g (*Hack et al., 1994*; *Rutter, 1995*), most of them are born preterm. Growing evidence has shown that the incidence and development of PTB is a complex process that is influenced by a variety of environmental and genetic factors (*He et al., 2017*; *Liu et al., 2017*; *Qiu et al., 2017*; *Xiao et al., 2016*). Therefore, it is helpful to develop approaches of effective prevention and treatment of neonatal morbidity and mortality by elucidating etiological factors contributing to PTB or LBW.

Recent results provide support to pre-pregnancy maternal body mass index (BMI) is one of the potential risk factors for PTB (*Hendler et al., 2005*; *Lynch et al., 2014*; *Shaw et al., 2014*) and LBW (*Han et al., 2011*; *Liu et al., 2016*). In recent decades, the prevalence of obesity and overweight among women in many countries has increased at an alarming rate, especially in developing countries. Different countries, regions and incomes have different patterns of overweight and obesity that is more common among women in developing countries and men in developed countries (*Ng et al., 2014*). Moreover, epidemiological studies have been suggested that maternal overweight and obesity have been shown to be association with PTB (*Lynch et al., 2014*; *Shaw et al., 2014*; *Su et al., 2020*), LBW (*Rahman et al., 2015*) and adverse maternal health outcomes, such as gestational hypertension (*Santos et al., 2019*) and cesarean delivery (*PaidasTeefey et al., 2020*; *Rahman et al., 2015*). Similarly, several observational studies show that underweight women in pre-pregnancy is the major risk factor for LBW and PTB (*Ehrenberg et al., 2003*; *Madzia et al., 2020*; *Qu et al., 2019*). However, the conclusions of various studies on the correlation between pre-pregnancy BMI and PTB appear to be paradoxical. It was reported that the risk of PTB in women with pre-pregnancy high BMI was significantly increased (*Baeten, Bukusi & Lambe, 2001*;

*Chen, Chen & Hsu, 2020*; *Cnattingius et al., 1998*; *Cnattingius et al., 2013*; *Weiss et al., 2004*; *Zhou et al., 2019*), whereas other studies contradicted this result, suggesting that women with a high pre-pregnancy BMI could have a protective impact on PTB (*Chen et al., 2013*; *Khashan & Kenny, 2009*; *Sebire et al., 2001*). Furthermore, similar results have also been reported on association between pre-pregnancy BMI and LBW (*Li et al., 2020*; *Ronnenberg et al., 2003a*; *Wu et al., 2020*). However, there are relatively few studies of the effects of pre-pregnancy BMI on subsequent pregnancies for first-time mothers among Southern Chinese women. Although several researchers (*Liu et al., 2019*; *Pan et al., 2016*; *Ronnenberg et al., 2003b*) have conducted relevant research in domestic, these previous papers are mainly limited to earlier data or pregnant women in rural areas. Thus, studies on the role of pre-pregnancy BMI in adverse pregnancy outcomes for first-time mothers in urban areas of Southern China remain scarce.

Given above controversial results, we conducted a hospital-based case-control study to determine whether there is a higher risk of adverse outcome in women with abnormal BMI as compared with normal BMI women among first-time mothers in urban areas of southern China.

## METHODS

### Study subjects

This was a hospital-based, case-control study conducted in The Third Affiliated Hospital of Guangzhou Medical University, Guangdong, China. Pregnant women who gave birth for the first time were included between January 2015 and December 2015. A total of 322 women with PTB and 3,362 women with term delivery controls were enrolled. These people also could be divided into 317 cases of LBW and 3,367 controls of normal birth weight. This study was approved by the institutional review board of The Third Affiliated Hospital of Guangzhou Medical University, Guangdong, China (Medical Ethics Hearing [2020]No.036). And Institutional Review Board waived the need for consent.

The preterm group was defined to deliver within 37 weeks of conception without congenital abnormalities or neurological damage. The control group between 37 and 42 weeks of gestation without congenital abnormalities or neurological damage was matched with the case in the residential area for one week at the same hospital. For both groups, we excluded women with a multiple pregnancy, stillbirth, prior deliveries, embryo transfer and in vitro fertilization.

All data for this study was collected from women who gave their first birth during hospitalization, including maternal age, education, gravidity, occupation, health insurance, height and weight before of pregnancy. Gestational hypertension, caesarean delivery, birth weight and gestational week at delivery were obtained from the medical records. Pre-pregnancy BMI defined as the body weight in kilograms divided by the square of the height in metres ($kg/m^2$). BMI is classified as underweight ($<18.5\ kg/m^2$), normal weight ($18.5–23.9\ kg/m^2$), overweight ($24.0–27.9\ kg/m^2$), and obesity ($\geq28.0\ kg/m^2$) according to the weight standard of Chinese adults (*Zhou, 2002*). In this study, child birth weight below 2500, g was considered LBW. Once the two occasions systolic blood pressure (BP)

or diastolic BP values of pregnant women measured at intervals of 24 h exceed 140 mmHg or 90 mmHg respectively, they are diagnosed as gestational hypertension.

## Statistical analysis

$t$ test (for continuous variables) or $\chi 2$ test (for categorical variables) was used to assess the difference in demographic characteristics between two groups. Adjusted odds ratios (OR) with 95% confidence intervals obtained from logistic regression model were used to quantify associations between pre-pregnancy BMI and adverse outcomes, adjusting for age, occupation, health insurance and education. We also further divided gestational age into four subtypes: extremely PTB (<28 gestational week), very PTB (28–31 gestational week), moderate PTB (32–36 gestational week) and normal (≥37 gestational week). Besides, we referred to the method provided by *Marchant et al. (2012)* apply to weight for gestational age, details were described as below: we combined birth weight and gestational age to dichotomise as being small for gestational age (SGA, less than the tenth percentile of weight for gestation) (*Oken et al., 2003*) and non-small for gestational age (NSGA, large than the tenth percentile of weight for gestation). As in previous studies, gestational week was classified into categories of term, 34–36 week and below 34 week. We then divided newborns into six groups: (1) term and NSGA; (2) 34–36 week gestation and NSGA; (3) below 34 week gestation and NSGA; (4) term and SAG; (5) 34–36 week gestation and SAG; (6) below 34 week gestation and SAG. A two-sided $p$-value of 0.05 or less was accepted to be statistically significant. Data were analyzed using Statistical Analysis Software 9.4. (SAS Institute, Cary, NC).

## RESULTS

### Characteristics of the study subjects

The baseline maternal characteristics are shown in Table 1. Of the 3,684 live births, 8.7% ($n = 322$) were PTBs, and the remaining 91.3% ($n = 3,362$) were term births. The proportions of underweight, normal weight, overweight, and obesity women were 23.29%, 63.36%, 10.72%, and 2.63%, respectively. There were significant differences between PTB group and terms group with respect to age, health insurance, occupation and education (all $P < 0.01$). There were no significant differences between the two groups with regard to gravidity ($P > 0.05$). The variables in the normal and the LBW group were basically the same as those in the term and preterm groups.

### Association analysis between pre-pregnancy BMI and adverse outcomes

The association between pre-pregnancy BMI and the risk of adverse outcomes are considered in Table 2 and Table 3. Underweight, overweight and obesity did not increase the risk of PTB as compare with normal weight (AOR = 1.01, 95% CI [0.76–1.34]; AOR = 1.25, 95% CI [0.87–1.80]; and AOR = 1.27, 95% CI [0.65–2.51], respectively). However, When PTB was stratified into three subgroups, both maternal overweight and obesity increased the risks of extremely PTB (AOR = 8.12, 95% CI [1.11–59.44]; AOR = 15.06, 95% CI [1.32–172.13], respectively).

**Table 1  Baseline maternal characteristics of the first-time mothers between two groups.**

| variable | Terms (N = 3362) | PTB (N = 322) | P | Normal (N = 3367) | LBW (N = 317) | P |
|---|---|---|---|---|---|---|
| Age | | | <0.001 | | | <0.001 |
|   <25 | 421 (12.52) | 55 (17.08) | | 418 (12.41) | 58 (18.30) | |
|   25–34 | 2628 (78.17) | 220 (68.32) | | 2630 (78.11) | 218 (68.77) | |
|   ≧35 | 313 (9.31) | 47 (14.60) | | 319 (9.48) | 41 (12.93) | |
| Health insurance | | | <0.001 | | | <0.001 |
|   Care for urban employees | 2269 (67.49) | 161 (50.00) | | 2261 (67.15) | 169 (53.31) | |
|   Free medical service | 87 (2.59) | 6 (1.86) | | 90 (2.67) | 3 (0.95) | |
|   Full-cost | 1006 (29.92) | 155 (47.14) | | 1016 (30.18) | 145 (45.74) | |
| Occupation | | | 0.004 | | | 0.007 |
|   Professional | 1534 (45.63) | 119 (36.96) | | 1532 (45.50) | 121 (38.17) | |
|   Business | 355 (10.56) | 28 (8.70) | | 358 (10.63) | 25 (7.89) | |
|   Housewife | 389 (11.57) | 46 (14.29) | | 388 (11.52) | 47 (14.83) | |
|   Others | 1084 (32.24) | 129 (40.06) | | 1089 (32.35) | 124 (39.12) | |
| Gravidity | | | 0.214 | | | 0.875 |
|   <2 | 2245 (66.78) | 204 (63.35) | | 2237 (66.44) | 212 (66.88) | |
|   ≧2 | 1117 (33.22) | 118 (36.65) | | 1130 (33.56) | 105 (33.12) | |
| Education | | | <0.001 | | | <0.001 |
|   Less than high school | 439 (13.06) | 86 (26.71) | | 444 (13.19) | 81 (25.55) | |
|   High school | 444 (13.21) | 38 (11.80) | | 443 (13.16) | 39 (12.30) | |
|   College | 2479 (73.73) | 198 (61.49) | | 2480 (73.65) | 197 (62.15) | |

**Notes.**

Abbreviation: PTB, preterm birth; LBW, low birth weight.

Pre-pregnancy underweight was significantly associated with the increased risk for LBW. In comparison with women who had normal pre-pregnancy BMI, women with low BMI category was more likely to deliver a LBW infant (crude OR=1.48, 95% CI [1.14–1.93]). After the adjustment for potential confounding factors, the AOR associated with the risk for giving birth to a LBW infant were 1.44 (95% CI [1.11–1.89]), the similar result was seen in Table 4 as compared with term and NSAG (AOR = 1.78, 95% CI [1.45–2.17]). Underweight was also significantly associated with a lower risk of gestational hypertension (AOR = 0.45, 95% CI [0.25–0.82]) and caesarean delivery (AOR = 0.74, 95% CI [0.62–0.90]). Both maternal overweight and obesity were found to be a risk factor for gestational hypertension (AOR = 1.71, 95% CI [1.06–2.77]; AOR = 5.54, 95% CI [3.02–10.17], respectively) and caesarean delivery (AOR = 1.91, 95% CI [1.53–2.38]; AOR = 1.85, 95% CI [1.21–2.82], respectively).

## DISCUSSION

Our study demonstrated that maternal underweight prior to pregnancy, as compared with normal weight women, significantly elevated the risk for LBW for the first-time mothers among Southern Chinese. Maternal underweight were also found to be at lower risk of gestational hypertension and caesarean delivery. In women who had a high pre-pregnancy

**Table 2 Crude and AOR for the association between pre-pregnancy BMI and adverse outcomes.**

| Outcomes | BMI status | Case | Control | OR (95% CI) | P | AOR* (95% CI) | P |
|---|---|---|---|---|---|---|---|
| PTB | Underweight | 75 (23.29) | 783 (23.29) | 1.04 (0.79–1.37) | 0.787 | 1.01 (0.76–1.34) | 0.943 |
| | Normal weight | 197 (61.18) | 2137 (63.56) | 1.00 (reference) | | 1.00 (reference) | |
| | Overweight | 40 (12.42) | 355 (10.56) | 1.22 (0.86–1.75) | 0.272 | 1.25 (0.87–1.80) | 0.224 |
| | Obesity | 10 (3.11) | 87 (2.59) | 1.25 (0.64–2.44) | 0.519 | 1.27 (0.65–2.51) | 0.486 |
| LBW | Underweight | 94 (29.65) | 764 (22.69) | 1.48 (1.14–1.93) | 0.003 | 1.44 (1.11–1.89) | 0.007 |
| | Normal weight | 179 (56.47) | 2155 (64.00) | 1.00 (reference) | | 1.00 (reference) | |
| | Overweight | 34 (10.73) | 361 (10.73) | 1.13 (0.77–1.66) | 0.521 | 1.17 (0.80–1.73) | 0.423 |
| | Obesity | 10 (3.15) | 87 (2.58) | 1.38 (0.71–2.71) | 0.343 | 1.41 (0.72–2.78) | 0.322 |
| Gestational hypertension | Underweight | 13 (9.92) | 845 (23.78) | 0.43 (0.24–0.78) | 0.006 | 0.45 (0.25–0.82) | 0.009 |
| | Normal weight | 80 (61.07) | 2254 (63.44) | 1.00 (reference) | | 1.00 (reference) | |
| | Overweight | 23 (17.56) | 372 (10.47) | 1.74 (1.08–2.81) | 0.022 | 1.71 (1.06–2.77) | 0.03 |
| | Obesity | 15 (11.45) | 82 (2.31) | 5.15 (2.85–9.33) | <0.001 | 5.54 (3.02–10.17) | <0.001 |
| Cesarean delivery | Underweight | 194 (17.60) | 664 (25.72) | 0.70 (0.59–0.84) | <0.001 | 0.74 (0.62–0.90) | 0.002 |
| | Normal weight | 686 (62.25) | 1684 (63.83) | 1.00 (reference) | | 1.00 (reference) | |
| | Overweight | 180 (16.33) | 215 (8.33) | 2.01 (1.62–2.50) | <0.001 | 1.91 (1.53–2.38) | <0.001 |
| | Obesity | 42 (3.81) | 55 (2.13) | 1.84 (1.22–2.77) | 0.004 | 1.85 (1.21–2.82) | 0.004 |

**Notes.**
Abbreviations: OR, odds ratio; CI, confidence interval; AOR, adjusted odds ratio; PTB, preterm birth; LBW, low birth weight.
*Adjusted OR and 95% CI were calculated by the logistic regression model after adjusting for age, health insurance, occupation and education.

**Table 3 Adjusted* a relative risk for the associations between pre-pregnancy BMI and PTB by gestational age.**

| Gestational age | Underweight | Normal weight | Overweight | Obesity |
|---|---|---|---|---|
| Term | – | 1.00 (reference) | – | – |
| Moderately PTB | 0.97 (0.71–1.33) | 1.00 (reference) | 1.18 (0.79–1.77) | 1.23 (0.58–2.59) |
| Very PTB | 1.05 (0.54–2.02) | 1.00 (reference) | 1.28 (0.55–2.96) | 0.77 (0.10–5.77) |
| Extremely PTB | 3.22 (0.53–19.59) | 1.00 (reference) | 8.12 (1.11–59.44) | 15.06 (1.32–172.13) |

**Notes.**
Abbreviations: PTB, preterm birth.
*Adjusted OR and 95% CI were calculated by the logistic regression model after adjusting for age, health insurance, occupation and education.

BMI, our study showed a significantly higher risk of gestational hypertension, caesarean delivery and extremely PTB.

The proportions of overweight and obesity were lower than underweight in our study, which were similar to previously reported data from other Chinese studies (*Pan et al., 2016*). Our finding showed that underweight women increase the risk of LBW in subsequent pregnancy, which is consistent with the result of a review study by *Liu et al. (2016)*. The study, including 60 studies covering 1,392,799 women, showed that infants had a higher risk of having a LBW when their mothers were underweight (OR, 1.67, 95% CI [1.39–2.02]) as compare to women with normal weight. Previous studies have demonstrated that maternal nutrition during pregnancy has great influence on providing the essential nutrients for fetal growth (*Nnam, 2015*) and pregnant women who are undernourished tend to have LBW infant (*Allen, 2001*). Maternal nutritional imbalance may be a key factor in the reduction of

**Table 4 Adjusted[*] a relative risk for the associations between pre-pregnancy BMI and weight for gestational age.**

| Weight for gestational age | Underweight | Normal weight | Overweight | Obesity |
|---|---|---|---|---|
| NSAG | | | | |
| Term | – | 1.00 (reference) | – | – |
| 34–36 week | 0.88 (0.58–1.34) | 1.00 (reference) | 1.29 (0.80–2.06) | 1.40 (0.59-3.29) |
| <34 week | 1.52 (0.94–2.47) | 1.00 (reference) | 1.48 (0.77–2.84) | 0.97 (0.23-4.11) |
| SAG | | | | |
| Term | **1.78 (1.45-2.17)** | 1.00 (reference) | 0.88 (0.63–1.21) | 1.89 (0.48-1.66) |
| 34–36 week | 1.49 (0.79–2.81) | 1.00 (reference) | 0.43 (0.10–1.82) | 0.86 (0.12-6.41) |
| <34 week | 0.73 (0.20–2.63) | 1.00 (reference) | 1.62 (0.44–5.89) | 2.23 (0.28-17.91) |

**Notes.**

Abbreviations: NSAG, non-small for gestational age; SAG, small for gestational age.

[*]Adjusted OR and 95% CI were calculated by the logistic regression model after adjusting for age, health insurance, occupation and education.

surface area and placental weight. In the lower surface area and placental weight, nutrient and waste transfer between the maternal and fetal circulation is reduced and other normal processes, such as fetal development and growth, are also restricted (*Lechtig et al., 1975*). Thus, maternal malnutrition may lead to LBW infant. Moreover, the finding in our study on association between underweight and LBW was also in line with previous literature (*Li et al., 2013*; *Liu et al., 2012*).

Previous literature has revealed that maternal overweight and obesity were associated with the increased risk for gestational hypertension and caesarian delivery (*Lewandowska, Wieckowska & Sajdak, 2020*; *Machado, Monteiro & Oliveira, 2020*; *Vince et al., 2020*). Our study also confirmed these findings. However, the risk of gestational hypertension and caesarean delivery were reduced among underweight mothers. Although mechanism by which obesity responsible for the increased risk of gestational hypertension or caesarean delivery is unclear, maternal obesity lead to an increase in the number and size of adipocytes or pelvic malacia. A significant amount of adipocytes has been proposed as a cause of excessive inflammatory reaction, pregnant female possibly experienced obstructive dystocia due to pelvic malacia lead to a relatively narrow birth canal. Therefore, both gestational hypertension and caesarean delivery were affected by obesity in obese mothers (*Kriketos et al., 2004*; *Young & Woodmansee, 2002*).

In our study, the risk of PTB is relatively higher among overweight and obese mothers but the association was not statistically significant (Table 2). However, this association has dramatically changed in a subgroup analysis of gestational age (Table 3). Our result regarding extremely PTB is consistent with a previous study (*Su et al., 2020*). Overweight and obesity are generally considered to be the risk factors for PTB due to the effects of placental insufficiency (*Lassance et al., 2015*; *Pereira et al., 2015*), inflammatory state (*Gaillard et al., 2016*), insulin sensitivity (*Catalano et al., 1999*) and cellular oxidative stress (*Ballesteros-Guzman et al., 2019*). However, conclusions on the relationship between pre-pregnancy BMI and PTB seem to be paradoxical among different studies. The differences emerged between studies could be attributed to study design or power, recall bias, multiple

comparisons, eating habits and different ethnicities. Additionally, the category of BMI was different among the studies.

Although some confounding factors had been controlled, alcohol consumption and maternal smoking were not adjusted as only one woman claimed to have a history of alcohol consumption and smoking in our data. So we did not adjust for these two variables.

However, there are a number of potential limitations of this study that merit consideration. One limitation of the current study is that it is difficult to distinguish between spontaneous and iatrogenic PTB, which we could not assess the association between special types of PTB and pre-pregnancy BMI. Additionally, the BMI we obtained in our study derived from weight and height information by women self-reported, which could lead to bias risk estimates of PTB (*Michels, Greenland & Rosner, 1998*).

## CONCLUSIONS

In conclusion, our study suggested that maternal overweight and obesity were associated with a significantly higher risk of gestational hypertension, caesarean delivery and extremely PTB. Underweight was correlated with an increased risk of LBW and conferred a protective effect regarding the risk for gestational hypertension and caesarean delivery for the first-time mothers among Southern Chinese.

## ACKNOWLEDGEMENTS

The authors are grateful to all subjects who participated in this study.

### Funding
The authors received no funding for this work.

### Competing Interests
The authors declare there are no competing interests.

### Author Contributions
- Li Li conceived and designed the experiments, performed the experiments, authored or reviewed drafts of the paper, and approved the final draft.
- Yanhong Chen and Li Ke conceived and designed the experiments, authored or reviewed drafts of the paper, and approved the final draft.
- Zhifeng Lin, Weiyan Lin and Weilin Ou performed the experiments, analyzed the data, prepared figures and/or tables, and approved the final draft.
- Yangqi Liu and Chengli Zeng analyzed the data, prepared figures and/or tables, and approved the final draft.

### Ethics
The following information was supplied relating to ethical approvals (i.e., approving body and any reference numbers):

The Ethics Committee of The Third Affiliated Hospital of Guangzhou Medical University approved this study (Medical Ethics Hearing [2020]No.036).

## Data Availability

The raw measurements are available in the Supplementary Files.

## Supplemental Information

Supplemental information for this article can be found online at http://dx.doi.org/10.7717/peerj.10123#supplemental-information.

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
