# Peer review of "Association of pre-pregnancy body mass index with adverse pregnancy outcome among first-time mothers"

_PeerJ, doi:10.7717/peerj.10123_

## Round 0.1 · original submission · Major Revisions

· Academic Editor

Major Revisions

You will find that both reviewers raised major concerns about your manuscript. They ask you not only to improve the language, background introduction, and proper citation of previous publications but also to improve the experimental design and explain why this study did not find an association between high BMI and PTB.

Reviewer 1 ·

Basic reporting

In this manuscript, Li Li et al conducted a study on a total of 3,684 women who gave birth for the first time in south of China to investigate the associations of preterm birth (PTB), low birth weight (LBW) with pre-pregnancy Body Mass Index (BMI). Their study suggests that underweight women increase the risk of LBW.
Although their result is consistent with many others, there are several major concerns as outlined below:

1. No big differences/advantages of this study are from the literatures (J Nutr. 2003 Nov;133(11):3449-55; BMC Pregnancy Childbirth. 2019 Mar 29;19(1):105....).

2. This manuscript should be clearer, more succinct, and professional. For example: in Abstract, "Risks of PTB and LBW in the first subsequent pregnancy in women who had a abnomal BMI were compared with risks in women who had a ideal BMI.”

3. It is better to show some data in Figure although the tables are accepted.

Experimental design

No comment

Validity of the findings

No comment

Additional comments

No comment

Reviewer 2 ·

Basic reporting

This study was designed to elucidate the association between maternal BMI and adverse pregnancy outcomes for first-time mothers in Southern China. The topic is valuable taking into account that obesity is increasing rapidly worldwide, particularly in rapid developing countries like China. However, I have some comments/concerns:

There have been many studies regarding the association of maternal BMI and offspring birth outcomes, in and out of China, such as Rahman MM, Abe SK, et al. 10.1111/obr.12293, Pan Y, Zhang S et al, 10.1136/bmjopen-2016-011227. The authors did not provide a comprehensive background in Introduction. Several latest published papers were not incorporated.

Moreover, the language is hard to follow. There are flaws and mistakes in written English. For ex, Introduction, line 58 “both” should be not capitalized; line 62, “is” should be “was”; Materials and methods, line 81 and 83, repeated “the control group”; Discussion Line 157, “showed” is in fact “showing”, line 158, “ruduced” should be “reduced”. It would be useful to have a native English speaker edit the manuscript for clarity and concise language.

Experimental design

1) First, the meaning of this study is not well explained in Introduction. Given the many reports on this topic, the authors did not address clearly the importance and novelty for this study. Is it for Southern Chinese, or first-time mothers?

2) As the authors mentioned, preterm birth is related to a variety of factors. Gestational hypertension may be a mediator between maternal BMI and PTB or low birth weight. To simply adjust for gestational hypertension in the regression is not suitable.

3) Gestational diabetes, twin pregnancy, and gestational weight gain are also important factors influencing the association. Given your dataset, are you able to analyze the association taking into account these factors?

4) Birth weight is a key indicator for intrauterine environment. However, birth weight adjusted for gestational age would be more appropriate than birth weight itself.

5) Also, information on early and late PTB is useful for health professionals. It would be useful to examine the risk of PTB according to gestational age with maternal BMI strata.

Validity of the findings

1) Table 2 and 3 are not well structured. In Results, the associations between overweight and obese mothers with offspring birth outcomes are not presented.
2) Please explain why this study did not find an association between high BMI and PTB.
3) The discussion part is not well organized and hard to follow.

---

## Round 0.2 · Minor Revisions

· Academic Editor

Minor Revisions

The revised manuscript has been improved, but please further improve the manuscript taking consideration of the reviewer's comments that are easy to address.

Reviewer 1 ·

Basic reporting

The revision improved the paper as written. There are several issues although the authors addressed most of concerns and suggestions in the revision.

1. Word document of the revised manuscript is not the same version of the PDF
2. One sentence is duplicated with the next one (line 221-222)
3. Grammatical issues will need to be addressed.
For example, line85-90; line131, line 132, line 162, line 178-179, line 212

Experimental design

What is the logical explanation for dividing newborns into six groups, not 8 groups? The authors divided gestational ages into four subtypes: extremely PTB (<28 gestational week), very PTB (28–31 gestational week), moderate PTB (32–36 gestational week) and normal (≧37 gestational week), and categorized newborn weight for gestational age into two groups: SGA and NSGA.

Validity of the findings

no comment

Additional comments

no comment

---

## Round 0.3 · accepted · Accept

· Academic Editor

Accept

The manuscript has been improved.